# Blood Transcriptomics Identifies Multiple Gene Expression Pathways Associated with the Clinical Efficacy of *Hymenoptera* Venom Immunotherapy

**DOI:** 10.3390/ijms25063499

**Published:** 2024-03-20

**Authors:** Ajda Demšar Luzar, Peter Korošec, Mitja Košnik, Mihaela Zidarn, Matija Rijavec

**Affiliations:** 1Laboratory for Clinical Immunology and Molecular Genetics, University Clinic of Respiratory and Allergic Diseases Golnik, 4204 Golnik, Slovenia; ajda.demsarluzar@klinika-golnik.si (A.D.L.); peter.korosec@klinika-golnik.si (P.K.); mitja.kosnik@klinika-golnik.si (M.K.); mihaela.zidarn@klinika-golnik.si (M.Z.); 2Biotechnical Faculty, University of Ljubljana, 1000 Ljubljana, Slovenia; 3Faculty of Pharmacy, University of Ljubljana, 1000 Ljubljana, Slovenia; 4Faculty of Medicine, University of Maribor, 2000 Maribor, Slovenia; 5Faculty of Medicine, University of Ljubljana, 1000 Ljubljana, Slovenia

**Keywords:** *Hymenoptera* venom immunotherapy, longitudinal transcriptomic profiling, tolerance induction, successful venom immunotherapy

## Abstract

Allergen-specific venom immunotherapy (VIT) is a well-established therapy for *Hymenoptera* venom allergy (HVA). However, the precise mechanism underlying its clinical effect remains uncertain. Our study aimed to identify the molecular mechanisms associated with VIT efficiency. We prospectively included 19 patients with HVA undergoing VIT (sampled before the beginning of VIT, after reaching the maintenance dose, one year after finishing VIT, and after a sting challenge) and 9 healthy controls. RNA sequencing of whole blood was performed on an Illumina sequencing platform. Longitudinal transcriptomic profiling revealed the importance of the inhibition of the NFκB pathway and the downregulation of *DUX4* transcripts for the early protection and induction of tolerance after finishing VIT. Furthermore, successful treatment was associated with inhibiting Th2, Th17, and macrophage alternative signalling pathways in synergy with the inhibition of the PPAR pathway and further silencing of the Th2 response. The immune system became activated when reaching the maintenance dose and was suppressed after finishing VIT. Finally, successful VIT restores the immune system’s balance to a state similar to that of healthy individuals. Our results underline the important role of the inhibition of four pathways in the clinical effect of VIT: Th2, Th17, NFκB, and macrophage signalling. Two biomarkers specific for successful VIT, regardless of the time of sampling, were *C4BPA* and *RPS10-NUDT3* and should be further tested as potential biomarkers.

## 1. Introduction

*Hymenoptera* venom allergy (HVA) is an IgE-mediated hypersensitivity reaction following a honeybee, vespid, or ant sting [1,2]. An allergic reaction to *Hymenoptera* species presents with severe manifestations and is potentially life-threatening [3]. HVA varies from large local reactions at the sting site to systemic reactions [4]. Systemic reactions vary significantly in severity, from moderate reactions consisting of generalised skin symptoms to severe life-threatening anaphylactic reactions affecting the cardiac and respiratory systems. The prevalence of systemic reactions is estimated to be up to 7.5% in adults and up to 3.4% in children, with anaphylaxis in 0.3–42.8% of cases [2,5,6]. The incidence of anaphylaxis has been increasing over the years, and its mortality rate is estimated to be between 0.03 and 0.48 per million population per year [7]. However, mortality may have been underestimated due to unrecognised stings in unexplained causes of death [8,9].

The only treatment is allergen-specific venom immunotherapy (VIT), which is reported to be effective in 77–84% of patients treated with honeybee venom and in 91–96% of patients treated with vespid venom [2,10]. Immunotherapy aims to restore immune tolerance and thus eliminate systemic allergic reactions after insect stings [2]. In contrast to aero and food immunotherapies, the immune tolerance established during VIT is considered to be lifelong, even after the discontinuation of treatment [11]. However, VIT is a long-lasting treatment, and in some cases, it should not be discontinued. Patients with known risk factors, namely, an initial severe sting reaction, clonal mast cell disorders, and/or elevated basal serum tryptase (>11.4 µg/L) [2,12,13,14], should receive prolonged or even lifelong treatment.

Many studies have focused on detecting biomarkers for VIT efficiency evaluation. Currently, laboratory parameters used as diagnostic methods to detect sensitisation are specific IgE (sIgE) levels and skin prick tests. However, there is no evidence that monitoring sIgE levels can predict the success of therapy [2]. Furthermore, IgG4 levels were studied. Its levels can be used for an evaluation of immunological responsiveness to VIT. It was suggested that not levels but rather functional activity might correlate with VIT effectiveness [15,16]. Previous studies evaluated the activation of effector cells using the basophil activation test and concluded that successful VIT decreases test sensitivity without changing the reactivity [17,18,19]. Other studies have focused on using serum CD30 levels to indicate a Th2 immune response. A decrease in serum CD30 levels may be considered an effective prognostic factor [20,21]. Most recent studies have highlighted the importance of the type of allergen the patient is sensitised to. Predominant sensitisation to the allergen Api m 1 showed no increased risk of treatment failure, whereas for patients predominantly sensitised to the allergen Api m 10, the risk of VIT failure increased. Therefore, molecular diagnostics has promising potential for the risk stratification of VIT [22,23,24]. But to date, no biomarkers have been established in clinical practice, making a controlled sting challenge the gold standard for evaluating venom tolerance, indicating clinical protection [2,25]. 

With the progress of science, the rapidly evolving field of genetics could be used to explore mechanistic changes in patients undergoing VIT and further propose potential biomarkers. Only a few transcriptome studies have been conducted in the research field of VIT. One of the first ones was performed by Tsicopoulus A., who investigated the mRNA profiles of *IL-4* and *IFNγ* in enriched T lymphocytes. A significant decrease in *IL-4* mRNA expression was observed after 90 days of VIT. Similarly, an increase in IFNγ expression was detected during VIT [26]. Later, in 2006, a microarray analysis of differentially expressed genes (DEGs) in peripheral blood mononuclear cells was performed. Three independent groups of patients were compared: the control group, patients after less than a year of VIT, and patients after five to six years of VIT. Osteopontin, a cytokine known to modulate Th1 activity, was significantly upregulated in patients after VIT [27,28]. In the following years, Niedoszytko et al. studied gene expression reflecting long-term protection in VIT. After comparing a group of patients with successful VIT to a group with unsuccessful VIT, a list of 18 genes forming part of a predictive model of successful VIT was proposed [29]. Furthermore, the same authors compared a group of patients before VIT with patients in the maintenance phase and with patients after VIT. Only 6 of 18 previously described genes were significantly expressed between the first and last aforementioned groups. Signature genes differentially expressed after reaching the maintenance phase of VIT, and signature DEGs still observed in patients after VIT were described [30]. Later, in 2018, a genome-wide analysis of expression before VIT and one year after VIT was conducted. It was the first study to include paired samples from the same patients during VIT. Although the findings did not align with previously described gene expression patterns, they were important, as they described the upregulation of leukocyte-specific genes expressed in antigen-presenting cells: macrophages and monocytes. The authors subsequently suggested that changes in T-cell subsets and the simultaneous restoration of the immune system’s innate branch may play a key role in establishing venom tolerance [31]. Over the last few years, an evolving method of single-cell RNA sequencing has been used for biomarker research [32,33,34], but not yet in the field of HVA or VIT.

Even though it is well documented that VIT provokes venom tolerance in most treated patients, the exact underlying mechanism remains unclear. We aimed to establish a less invasive method than the sting challenge to assess the effect of VIT. With follow-up transcriptome analysis of paired blood samples, we tried to contribute to the multi-level fingerprint for VIT monitoring and the evaluation of its effectiveness. By gaining new insights into underlying mechanisms, novel possibilities for biomarker research are uncovered. 

## 2. Results

### 2.1. Characterisation of Transcriptome Fingerprint Characteristic of Successful VIT

Of 19 patients enrolled, 14 tolerated exposure to the allergen at the sting challenge and were classified as successfully completing the treatment, while patients with recurrent systemic reaction to sting, evaluated as Ring and Messmer grading I-II, were classified as treatment failure. Interestingly, only one patient with successful VIT had complications during the build-up phase of VIT, whereas all five patients with subsequent treatment failure experienced adverse reactions during the beginning of treatment. 

We aimed to characterise the transcriptome fingerprint and its differences between patients who tolerated a sting after VIT and those with treatment failure, enabling a further understanding of the mechanisms underlying VIT and thus contributing to the search for potential biomarkers of VIT efficiency. To our knowledge, this is the first transcriptome analysis designed as a follow-up study with paired samples before VIT, after reaching the maintenance dose, one year after finishing VIT, and right after the following sting challenge. The outcome of transcriptome analysis is a large amount of data, which need to be processed thoroughly. Thus, biomarker selection filtration criteria and a comparison of selected DEGs with previously published data were necessary. Accordingly, we first performed a follow-up analysis at different time points of VIT, followed by a detailed examination of overlapping DEGs (Figure 1) and a comparison to previously proposed biomarkers. Since immune cells represent only 1% of all blood cells [35], the transcriptome changes regarding the immune system were moderate; therefore, a *p*-value < 0.05 was used as a filtration criterion instead of a stricter false discovery rate *p*-value. Combined with the stringent criterion of a fold change higher than five, we gained a strong forecast of the selected DEGs’ significance. A detailed list of DEGs characteristic of successful VIT after reaching the maintenance dose (maintenance dose vs. before; maintenance dose vs. after; 76 DEGs), DEGs specific for the maintenance dose and after VIT (maintenance dose vs. before; after vs. before; 91 DEGs), and DEGs specific only for the time point after the treatment (after vs. before; maintenance dose vs. after; 254 + 49 DEGs) are presented in Appendix A, respectively. DEGs overlapping with treatment failure (14) are unspecific and were excluded from further analysis. Further results are focused on patients with successful VIT.

### 2.2. The Immune System Seems to Be Activated after Reaching the Maintenance Dose and, Conversely, Significantly Suppressed after Finishing VIT

After the maintenance dose of 100 µg is reached, the patient is believed to be protected from subsequent *Hymenoptera* stings. The mechanism behind gained protection differs from the mechanisms acquired through the following years of VIT. DEGs (76) specific for the maintenance dose that are different from those at time points before and after immunotherapy are presented in Appendix A. We would like to highlight three upregulated immune-related DEGs: *SERPINB4, HLA-U*, and *IGLV2–5* (Table 1). Interestingly, the serin protease inhibitor encoded by the *SERPINB4* gene seems to be upregulated only at the time point of the maintenance dose. *SERPINB4* has been previously associated with early inflammation in atopic dermatitis and psoriasis [36]. Similarly, the upregulation of the human leukocyte antigen can be observed, which has been directly associated with immune stimulation against external antigens [37]. DEGs that describe the immune system characteristics established after reaching the maintenance dose and that remain the same even after finishing immunotherapy are mainly downregulated (75/91; 83%). DEGs closely related to the immune system (referred to as hub genes) are presented in Table 2. Overall, not only the proportions of down-/upregulated genes but also their functional characterisation suggest that the immune system is activated after reaching the maintenance dose of the treatment and suppressed later after finishing VIT. 

### 2.3. NFκB Pathway Inhibition and Downregulation of DUX4 Transcripts Are Important for Early Protection and Tolerance Induction

The two DEGs specifically downregulated at both time points (maintenance dose and after VIT) are the pseudogenes *DUX4L10* and *DUX4L12*. Furthermore, the *DUX4, DUX4L2,* and *DUX4L14* transcripts were significantly downregulated after finishing VIT. The *DUX4* gene has been previously investigated in the context of other diseases [39]. Its transcriptomic profiling revealed its involvement in inflammation, cellular migration, and chemotaxis [40]. The pseudogenes *DUX4L10* and *DUX4L12* were significantly downregulated after reaching the maintenance dose (FC = −240, *p*-value = 6.46 × 10^−7^; FC = −90, *p*-value = 5.48 × 10^−5^, respectively), suggesting the limitation of the inflammatory response. Interestingly, the *DUX4* gene was significantly downregulated only after finishing VIT (after vs. before; fold change = −18.5, *p*-value = 0.0007). 

The most significant DEG expressed as early as reaching the maintenance dose, which persisted even one year after finishing VIT, is the downregulated *IKBKGP1* transcript, influencing the NFкB pathway. NFкB represents a family of transcription factors that regulate the expression of hundreds of genes, most of which are involved in immune and inflammatory responses [41]. Its signalling is essential to effective innate and adaptive immune responses, playing critical roles in the development, survival, and activation of B lymphocytes. There are two pathways of NFкB signalling, namely, canonical and non-canonical [42]. In the canonical pathway, upon the stimulation of B-cell receptors, CD40, or Toll-like receptors, the IкB kinase (IKK) complex liberates NFкB dimers, which translocate to the nucleus, where they control the transcription of target genes. The NF-kappa-B essential modulator (also known as NEMO) is the regulatory subunit of the IKK complex. NEMO is encoded by the Inhibitor of the Nuclear Factor Kappa B Kinase Subunit Gamma (*IKBKG*) gene. Its pseudogene *IKBKGP1* is highly significant for patients who tolerate stings. It is important to note that the significance of pseudogenes is being discovered and has been highlighted in various studies [43,44,45]. The transcript *IKBKGP1* was 5 times less expressed after reaching the maintenance dose (*p*-value = 0.01) and 25 times less expressed after immunotherapy (*p*-value = 0.00001). Its significant downregulation after VIT suggests lower immune response activation. Comparing successful and treatment failure patients, no difference in the transcript level can be observed before the initiation of VIT. In contrast, its significant downregulation is characteristic of the group with successful VIT after treatment (Figure 2).

Interestingly, the connection between VIT and the NFкB pathway was highlighted in a study by Kempinski et al. [46]. According to their research, the only gene that changed significantly during the build-up phase of VIT was *COMMD8*, which influences IкB degradation and, thus, the release of NFкB for nuclear translocation [46,47]. Moreover, the downregulation of the *COMMD8* transcript was also a part of the prediction model of successful VIT in a study performed by Niedoszytko et al. [29], further highlighting the vital role of the NFкB pathway in mechanisms of allergen tolerance established during VIT.

### 2.4. Th2- and Th17-Related DEGs Are Significantly Downregulated after Successful Immunotherapy

Of 410 DEGs in the group with successful VIT, those closely related to the immune system are described in Table 3. Of these, 96 overlapped with the time point of the maintenance dose. Our analysis of follow-up samples’ differential expression patterns before VIT, after reaching the maintenance dose, and after finishing VIT in the group with successful VIT shows the apparent downregulation of Th2- and Th17-related immune response transcripts. Furthermore, only one-quarter of DEGs after immunotherapy were upregulated and are primarily involved in negative immune system regulation. Hence, we assume that immunotherapy does not promote specific immune system pathways but, conversely, downregulates excessive immune processes. After successfully finishing immunotherapy, DEGs involved in the Th2 immune pathway (*EBI3*, *UMOD*), the Th17 immune pathway (*PHBP6*), tryptophan metabolism (*TDO2*), the production of inflammatory mediators (*PLA2G10*), and the innate branch of the immune system (*CPN1*, *MMD2, NLRP8*) were significantly downregulated. Notably, the *PHBP6* transcript, a pseudogene of prohibitin (*PHB1*), was significantly downregulated after immunotherapy (after vs. before; fold change = −24, *p*-value = 0.001). Prohibitin is known to interact with *STAT3* to affect IL17 secretion in Th17 cells. It is also required to activate the NFκB signalling pathway [48]. The downregulation of the Th17 pathway seems to be one of the prevailing mechanisms important for establishing allergen tolerance. 

Furthermore, the important involvement of B lymphocytes can be observed through possible isotype switching by the upregulation of *IGKV2D-28,* which is involved in antigen recognition.

On the other hand, fewer changes are seen through transcriptome analysis in the group of patients with treatment failure. One of the downregulated genes is the *IFI44L* transcript, which is known to promote macrophage differentiation and facilitate inflammatory cytokine secretion. Furthermore, the *TRIL* transcript, a component of the TLR4 signalling complex that mediates the innate immune response, was significantly upregulated after immunotherapy. Taken together, VIT seems to impact both humoral and innate immune system pathways in both groups of patients, namely, successful treatment and treatment failure, although seemingly to a much lower extent in the group of patients with treatment failure.

### 2.5. Successful Venom Immunotherapy Seems to Restore the Balance of the Immune System to a State Similar to That of Healthy Individuals

Comparing the transcriptome signatures of patients with successful treatment or treatment failure with healthy controls, a higher proportion of similarity can be observed between healthy controls and patients with successful VIT after immunotherapy (Figure 3a,b). In the group with successful VIT, there were 1768 DEGs compared to healthy controls before the treatment, whereas there were only 476 DEGs after VIT. In contrast, in the group with treatment failure, there were 1745 DEGs before VIT and 1012 DEGs after VIT compared to healthy controls. Hence, patients tolerating stings have a more similar expression pattern to healthy controls than patients with treatment failure (Figure 3c). This observation was previously described by Niedoszytko et al. [30], who discovered that immunotherapy seems to re-establish a balance similar to that in non-allergic individuals by silencing the immune system. Also noteworthy is the comparison of transcriptome signatures between successful VIT and treatment failure before and after the treatment, which are relatively similar, with only slight significant differences. Nevertheless, the dissimilarity is more notable after the treatment than before VIT (Figure 3a,b). 

### 2.6. Two Biomarkers Specific for Successful VIT, Regardless of the Time of Sampling, Are C4BPA and RPS10-NUDT3

The most important outcome of differential gene expression between groups at different time points of VIT is the identification of two DEGs referred to as biomarkers of successful VIT. The two transcripts significantly expressed in the group with successful VIT were C4b-binding protein alpha chain transcript (*C4BPA*) and *RPS10-NUDT3*. The expression of the *C4BPA* transcript was 10 times higher before and after immunotherapy in comparison to healthy controls, and its expression was 14 times and 9 times higher before and after immunotherapy in comparison to treatment failure, respectively. Conversely, the *RPS10-NUDT3* transcript was significantly downregulated in comparison to healthy controls and those with treatment failure (Figure 3a). C4b-binding protein is a complement inhibitor of the uncontrolled activation of the classical and lectin complement pathways. Besides complement inhibition, it has been studied as a regulator of excessive inflammation in chronic diseases, like systemic lupus erythematosus, and in cancer [49]. To our knowledge, *C4BPA* and *RPS10-NUDT3* have not been connected to HVA or VIT before. 

### 2.7. Inhibition of Macrophage Alternative Signalling Pathway in Synergy with Inhibition of PPAR Pathway Results in Silencing of Th2 Immune Response Characteristic of Successful VIT

Many differences in active pathways between patients with successful VIT and patients with treatment failure can be seen when characterising DEGs. The major blood-related pathways before the initiation of VIT demonstrated the activation of IL-6 signalling, TREM-1 signalling, and macrophage alternative activation signalling pathways in successful VIT patients. In contrast, IL-10 signalling, CLEAR signalling, and PPAR signalling pathways seem to be inhibited compared to patients with treatment failure. After immunotherapy, pathways related to B lymphocytes appear to be the most enriched, namely, B-cell receptor signalling, PI3K signalling, and FcµRIIB signalling. Outstanding is the pathway activation of the FcµRIIB receptor expressed on B lymphocytes, the only inhibitory Fc receptor [50]. Its crosslinking with the B-cell receptor (BCR) results in the inhibition of activation, proliferation, antigen internalisation, and antibody secretion [51]. Furthermore, the immune response is silenced by PI3K signalling, which has been shown to inhibit mast cell degranulation [52]. 

The importance of macrophages and the subsequent involvement of the immune system’s innate branch in establishing venom tolerance was previously observed by Karpinski et al. [31]. Our results further support their theory. The macrophage alternative activation signalling pathway is shown to be inhibited in successful VIT after the treatment, alongside the peroxisome proliferator-activated receptor (PPAR) and retinoid X receptor signalling pathways. The canonical pathway of alternative macrophage activation signalling involves the activation of STAT6 by the Th2 cytokines IL-4 and IL-13. The transcriptional synergy between PPAR/retinoid X receptor regulators and STAT6 is known to sustain the immune effector response [53]. Several studies have highlighted the important role of the activated PPAR pathway in promoting the Th2 immune response associated with allergic disease [54,55,56]. The downregulation of macrophage activation and the downregulation of the PPAR pathway reflect a lower Th2 immune response. These results again emphasise the importance of downregulating the immune system and establishing allergen tolerance gained over the years of VIT.

### 2.8. Protein–Protein Interaction before and after Venom Immunotherapy Identifies Different Master Regulators

The protein–protein interaction network analysis of selected DEGs in successful VIT vs. treatment failure was performed using Ingenuity Pathway Analysis to identify master regulators and their concomitant causal networks. Before VIT, 39 master regulators were identified. Of these, 27 predicted activation, and 12 predicted inhibition. The causal network with the most target-connected regulators is the EGFR network (Appendix A). The *EGFR* regulator impacts the NFκB and STAT3 pathways, promoting an inflammatory response. After VIT, nine master regulators were identified. Of these, five were related to activation, and four predicted inhibition. The causal network with the most target-connected regulators is RASSF1 (Appendix A). The *RASSF1* master regulator significantly inhibits the *NFκB* subunit (*RELA*), further supporting the follow-up analysis suggesting that the significant inhibition of the NFκB pathway after VIT is characteristic of patients with successful VIT.

### 2.9. Different Mechanisms after Sting Challenge in Groups with Successful VIT and Treatment Failure

Differences in the immune response after a sting challenge between the group of patients tolerating the sting and the group of patients with treatment failure are evident. After the sting challenge, successful VIT establishes a tolerance to the allergen using different mechanisms, which can be seen in transcriptome signatures, as highly expressed DEGs in the group with treatment failure are completely downregulated in successful VIT (Figure 4). The major blood-related pathways in the group with successful VIT after the sting showed activated pathways of phagosome formation, the PI3K cascade, and potassium channels. The PI3K signalling cascade, inhibiting mast cell degranulation, is characteristic of patients with successful VIT tolerating a sting. Conversely, functional and pathway analysis in the group with treatment failure after the sting resulted in enriched pathways in B-cell receptor signalling, FCƐRI signalling, FCγR-dependent phagocytosis, and the binding and uptake of ligands by scavenger receptors. 

## 3. Discussion

We used transcriptome signatures to characterise differences between patients with successful VIT, who tolerated a sting challenge, and patients with treatment failure. Transcriptome analysis offers a bridge to the gap of known VIT mechanisms of gained allergen tolerance and opens new biomarker possibilities. Our results underline the important role of four immunological mechanisms: inhibition of Th2, Th17, and NFκB pathways and the involvement of macrophages in establishing venom tolerance; they also identify *C4BPA* and *RPS10-NUDT3* as possible biomarkers of successful VIT. 

To our knowledge, this is the first study to use follow-up samples before VIT, after reaching the maintenance dose, after finishing VIT, and after the sting challenge. Investigating patients’ transcriptome signatures, we outlined DEGs that were characteristic of patients who tolerated a sting in the controlled sting challenge. Since allergy and immunotherapy both consist of various complicated, interacting immune system mechanisms, we assumed that the specific dynamics of differentially expressed genes indicating different activated or inhibited pathways could be specific for different outcomes of VIT rather than a single measurement of a specific DEG. With this study being the first follow-up study on *Hymenoptera* venom immunotherapy, various previously described mechanisms have been confirmed through differential expression patterns. 

After reaching the maintenance dose, three transcripts, namely, *SERPINB4, HLA-U*, and *IGLV-2*, seem to describe the activated state of the immune system. *SERPINB4* has been previously associated with early inflammation in atopic dermatitis and psoriasis [36]. Similarly, human leukocyte antigen has been directly associated with immune stimulation against external antigens [37]. In contrast, DEGs that are significantly expressed after reaching the maintenance dose and maintained even after finishing VIT show significant inhibition of the immune system. Overall, the immune response seems to be induced after reaching the maintenance dose and, conversely, suppressed after finishing VIT. The *DUX4* gene is significantly downregulated after reaching the maintenance dose and after finishing VIT. Its transcriptomic profiling in previous studies revealed its involvement in inflammation, cellular migration, and chemotaxis [40]. The downregulation of the *DUX4* gene suggests a lower inflammatory state, which enables early protection and, later, allergen tolerance. Furthermore, the NFκB pathway is significantly inhibited. NFκB signalling is essential to effective innate and adaptive immune responses. It plays important roles in the development, survival, and activation of B lymphocytes [42]. The inhibition of the NFκB pathway through the downregulation of the *IKBKGP1* transcript suggests overall lower immune activation. The set of genes proposed as a prediction model for VIT evaluation, as described by Niedoszytko et al. [30], was analysed. However, of 18 proposed genes, only the *COMMD8* transcript indirectly coincides with our observation of the downregulation of the NFκB pathway, which furthermore highlights the vital role of the NFκB pathway in gaining early protection after reaching the maintenance dose and allergen tolerance after finishing VIT.

In the group of patients with successful VIT, a previously suggested shift from the Th2 to Th1 immune response [57] can be observed through a trend of primarily downregulated DEGs. In addition, the downregulation of DEGs related to the Th17 immune pathway (*PHBP6*), tryptophan metabolism (*TDO2*), the production of inflammatory mediators (*PLA2G10*), and innate immune system branches (*CPN1*, *MMD2*, *NLRP8*) is highly significant for successful VIT. According to our transcriptome analysis, the silencing of the Th2 immune response seems to be a consequence of the inhibition of the macrophage alternative signalling pathway in synergy with the inhibition of the PPAR pathway. The importance of the involvement of the innate immune system branch, specifically macrophage involvement, was previously highlighted by Karpinski et al. in the analysis of gene expression one year after VIT [31]. Furthermore, the inhibition of the Th17 pathway seems to be one of the prevailing mechanisms important for establishing allergen tolerance. *PHBP6*, a pseudogene of prohibitin, is highly downregulated. It is known to interact with *STAT3* to affect IL17 secretion in Th17 cells. Also important is the involvement of B lymphocytes, which can be observed through possible isotype switching by the upregulation of the *IGKV2D-28* transcript. *IGKV2D-28* is involved in antigen recognition.

It is important to state that most DEGs were downregulated after immunotherapy, as was similarly proposed by Konno et al. [27,28] and Niedoszytko et al. [29]. In contrast, Karpinski et al. [31], in a follow-up study, discovered a bigger proportion of upregulated genes one year after VIT. In our study, most of the DEGs were downregulated at both time points of VIT—after reaching the maintenance dose and after finishing VIT (maintenance dose: 385/486; 80%; after VIT: 317/410; 77%). Only roughly one-quarter of all DEGs after immunotherapy were upregulated and are primarily involved in negative immune system regulation. Hence, we assume that immunotherapy does not promote specific immune system pathways but, conversely, downregulates excessive immune processes.

Taken together, VIT seems to impact both humoral and innate immune systems in both groups of patients but seemingly does not result in immune tolerance in the group of patients with treatment failure. 

Comparing transcriptome signatures of the three groups—successful VIT, treatment failure, and healthy controls—we identified similar expression patterns between healthy controls and patients with successful VIT after immunotherapy. Immunotherapy seems to restore the balance of the immune system to a state similar to that of non-allergic individuals. Many differences between patients with successful VIT and those with treatment failure can be observed. The major blood-related pathways in successful VIT patients before the initiation of VIT demonstrated the activation of IL-6 signalling, TREM-1 signalling, and macrophage alternative activation signalling pathways and the inhibition of IL-10 signalling, CLEAR signalling, and PPAR signalling pathways. In contrast, after immunotherapy, pathways regarding B lymphocytes appear to be the most enriched, namely, B-cell receptor signalling, PI3K signalling, and FcµRIIB signalling. The most outstanding pathway after immunotherapy in patients with successful VIT is the activation of the FcµRIIB receptor expressed on B lymphocytes, which is the only inhibitory receptor. Its crosslinking with the BCR results in the inhibition of activation, proliferation, antigen internalisation, and antibody secretion [51]. Furthermore, the macrophage alternative activation signalling pathway is shown to be activated before VIT and, conversely, inhibited after VIT in the group with successful VIT. The canonical pathway of alternative macrophage activation signalling involves the activation of STAT6 by the Th2 cytokines IL-4 and IL-13. Additionally, after VIT, the PPAR and retinoid X receptor signalling pathways seem to be inhibited. The transcriptional synergy between PPAR/retinoid X receptor regulators and STAT6 is known to sustain the immune effector response. The inhibition of the macrophage activation signalling pathway and the inhibition of the PPAR pathway reflect a lower Th2 immune response. 

The important result of the transcriptome analysis of the follow-up samples is the identification of two biomarkers of successful VIT, the *C4BPA* and *RPS10-NUDT3* transcripts, that differentiate patients from those with treatment failure and from healthy controls before and after VIT. C4BPA acts as a complement inhibitor and has been studied as a regulator of excessive inflammation in chronic diseases [49]. To our knowledge, neither of the two transcripts has ever been associated with HVA or VIT before. Future confirmational studies on larger cohorts of patients are required to evaluate the potential of the two proposed biomarkers.

The previously proposed biomarkers of VIT efficiency (osteopontin, CD30, IL-4, IFNγ) were studied in detail and were not significantly expressed after reaching the maintenance dose or after finishing immunotherapy.

The transcriptome signature of the sting challenge showed clear differences in expression patterns between patients with successful VIT and those with treatment failure. Many more DEGs were upregulated in the group with successful VIT, but those silenced were shown to be significantly upregulated in the group of patients with treatment failure. Even though VIT suppresses the overall immune system in patients who tolerate the sting, it seems that upregulated DEGs also suppress inappropriate responses of the immune system, and those downregulated are responsible for allergic reactions. The major blood-related pathways in the group with successful VIT after the sting showed activated pathways of phagosome formation, PI3K cascade, and potassium channels. The PI3K cascade is known to inhibit mast cell degranulation, contributing to allergen tolerance.

The results of our analysis coincide with other studies performed in the field of immunotherapy to aero and food allergens (AIT), such as house dust mites, grass pollen, and eggs. Wide differences have been pointed out between VIT and AIT. To date, the mechanisms of hypersensitivity reactions that differentiate between respiratory and oral allergies from HVA have not been clarified. The most recent remarkable study by Huang et al. performed on house dust mite-allergic patients treated with AIT, similarly to our study, revealed the significant downregulation of the innate immune response, the downregulation of NFκB and PPAR pathways, and a lower involvement of Th17 signalling with subsequent decreases in Th17 and Th2 cytokines [58]. Furthermore, a single-cell RNA analysis on grass pollen-allergic patients demonstrated the downregulation of genes associated with Th2 signalling and the downregulation of antigen-presenting pathways in B lymphocytes [34]. Notably, a transcriptome study on children receiving oral immunotherapy for egg allergy also highlighted that the majority of DEGs after 8 months of AIT were downregulated, and some of them played an important role in IL-17 signalling [59]. Even though the differences between VIT and AIT for aero and food allergens have been pointed out, our results of the underlying mechanisms of gained allergen tolerance coincide and are very important for further research directions.

We want to point out that the potential limitation of our study is the fact that we used whole-blood mRNA for whole-transcriptome analysis. Since significant changes in gene expression in VIT are likely related to a small subset of cells, single-cell analysis would be reliable for the further evaluation of the presented results, as would polymerase chain reaction studies for the confirmation of identified gene fingerprints. Further, since the sting challenge is highly unpleasant and, in the case of treatment failure, also life-threatening for the patient, a possible limitation of the study is the small number of patients with treatment failure. An advantage of our study is the follow-up samples from the same patients monitored from the beginning of VIT until the end of treatment. Furthermore, the evaluation of VIT success was determined using the controlled sting challenge.

Taken together, our results underline the important role of four main known immunological mechanisms, namely, the inhibition of Th2, Th17, and NFκB pathways and the involvement of macrophages in establishing venom tolerance. We suggest further evaluating two possible biomarkers of successful VIT prognosis: *C4BPA* and *RPS10-NUDT3*. 

Since the controlled sting challenge is very unpleasant for HVA patients, future studies should strive to find a biomarker of VIT efficacy evaluated from patients’ blood. We suggest that a biomarker fingerprint rather than a single measurement could help predict VIT outcomes.

## 4. Materials and Methods

### 4.1. Study Group 

In this prospective study, we enrolled 19 patients with *Hymenoptera* venom allergy undergoing VIT at the University Clinic of Respiratory and Allergic Diseases Golnik (Slovenia) and 9 healthy controls with no diagnosed allergies. The study group characteristics are described in Table 4. According to the clinical history and skin and/or blood testing, honeybee VIT was indicated. Patients underwent honeybee ultra-rush VIT as previously described [2,25,60,61]. Whole-blood samples were collected before the beginning of VIT, after reaching the maintenance dose, one year after finishing VIT, and after a controlled sting challenge [2] (Figure 5). A sting challenge is still the most reliable method for evaluating VIT efficiency and the gold standard for clinicians. The sting challenge confirmed successful treatment in 14 patients, while 5 patients were considered as treatment failure. Treatment failure was characterised as Ring and Messmer grading I-II, and the reaction was immediately treated with epinephrine. The study was approved by the Slovenian National Medical Ethics Committee (KME 0120-443/2020/3). All patients gave their written informed consent to participate in the study.

### 4.2. Allergy Diagnostic Tests

In the laboratory evaluation, serum concentrations of specific honeybee IgE and total IgE were measured before the beginning of VIT on an Immulite (Siemens Healthcare GmbH, Erlangen, Germany). Sensitisation was defined as a sIgE level of 0.35 kIU/L or higher. The descriptive statistics are presented as medians and min–max ranges for measurement data and percentages for categorical data. Fisher Exact, Chi-square, or Mann–Whitney tests were used as appropriate to determine statistically significant differences using GraphPad Prism 10 software (version 10.2.1 for Windows; GraphPad Software, San Diego, CA, USA).

### 4.3. RNA Samples

Whole-blood samples containing an RNA-stabilising reagent were collected in PAXgene blood tubes (Qiagen, Hilden, Germany), which contain a proprietary reagent composition based on patented RNA stabilisation technology. Intracellular RNA was then isolated from human whole-blood leukocytes using the PAXgene Blood miRNA Kit on a fully automated QIAcube system (Qiagen) according to the manufacturer’s instructions. The integrity of RNA was determined using an RNA 6000 Nano LabChip kit (Agilent Technologies, Santa Clara, CA, USA) and an Agilent2100 Bioanalyser (Agilent Technologies). All samples with RIN integrity values greater than six were suitable for further sequencing.

### 4.4. RNA Sequencing and Bioinformatics Analysis

Sequencing libraries were prepared using the TruSeq Stranded TotalRNA Library workflow with Ribo-Zero-Globin (Illumina, San Diego, CA, USA). RNA sequencing (RNAseq) was performed on the Illumina NovaSeq6000 platform with 100 bp paired-end reads by Macrogen Inc. (Seoul, Republic of Korea). RNA sequencing data were then uploaded to the CLC Genomics Workbench (v.23.0.2, Qiagen Bioinformatics) [62]. Briefly, quality trimming to exclude low-quality reads, adapter trimming, alignment with *Homo Sapiens* (reference sequence_hg19, gene track_ensembl v87, mRNA track_ensembl v87), and global TMM normalisation of the reads were performed. The ENSEMBL gene annotation reference was used, which includes annotations of coding and non-coding genes (long non-coding RNAs, pseudogenes, small RNAs, and IG/TR transcripts) [63]. Genes with low expression (mean expression < 0.01) were filtered out. Differential gene expression was analysed within groups at different time points and between groups using the Wald test against the control group. Filtration criteria for differentially expressed genes were set to a *p*-value of less than 0.05 and a fold change greater than five. Similarly, filtration criteria to identify differentially expressed genes after the sting challenge were set to a *p*-value less than 0.05 and a fold change of more than two. Furthermore, a network and pathway analysis was performed using Ingenuity Pathway Analysis (IPA, Qiagen). 

## 5. Conclusions

The transcriptome analysis of follow-up samples from the same patients before VIT, after reaching the maintenance dose, one year after finishing VIT, and after a controlled sting challenge reveals the significant downregulation of DEGs related to the immune system. The significant inhibition of the Th2, Th17, and NFκB pathways is characteristic of successful VIT. Further evaluation of two possible biomarkers of successful VIT prognosis, namely, *C4BPA* and *RPS10-NUDT3*, is needed. Overall, the immune system seems to be activated after reaching the maintenance dose of the treatment and later suppressed after finishing VIT.

Since allergy is a complex disease, we suggest a biomarker fingerprint rather than a single biomarker, which could help predict VIT outcomes. Downregulated transcripts indicating lower immune activity and previously proposed biomarkers combined could provide a stronger forecast of VIT efficiency.

## Figures and Tables

**Figure 1 ijms-25-03499-f001:**
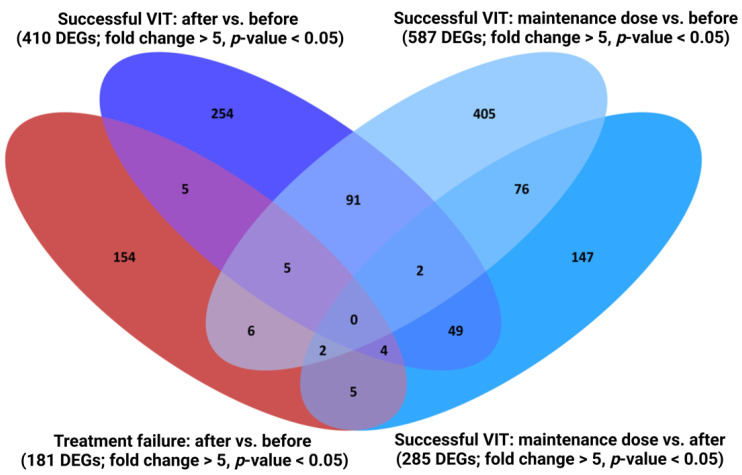
Follow-up analysis and overlapping of differentially expressed genes at three time points (before, after reaching maintenance dose, and after) for successful venom immunotherapy (blue) and at two time points (before and after) for treatment failure (red). (Cut-off: fold change > 5, *p*-value < 0.05.) VIT: venom immunotherapy; DEGs: differentially expressed genes.

**Figure 2 ijms-25-03499-f002:**
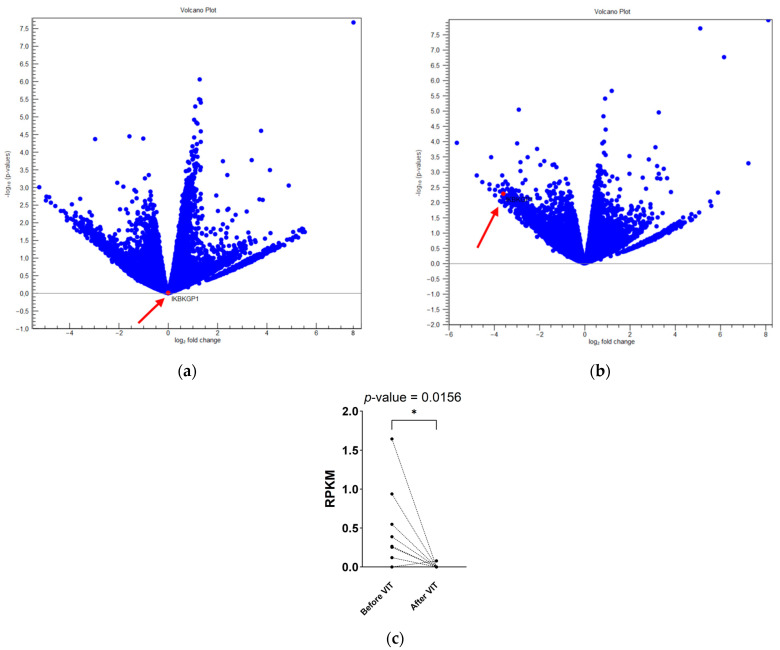
A volcano plot of differentially expressed genes between successful venom immunotherapy and treatment failure groups. Highlighted and with red arrows marked is the transcript of interest, *IKBKGP1*. No difference between the two groups before the initiation of venom immunotherapy is observed (**a**), whereas a significant difference between the two groups after venom immunotherapy is pointed out (**b**). The *IKBKGP1* transcript is significantly downregulated after venom immunotherapy in the group with successful treatment. The kinetics of the *IKBKGP1* transcripts of individual samples from successful venom immunotherapy are shown, with significant downregulation after venom immunotherapy (*p*-value = 0.0156) (**c**). RPKM: reads per kilobase per million mapped reads; *: statistical significance.

**Figure 3 ijms-25-03499-f003:**
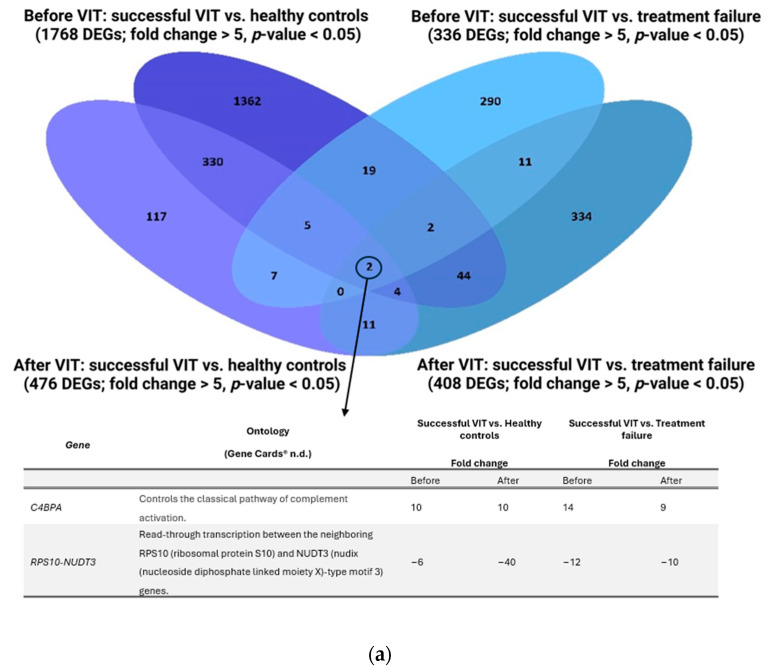
Comparison of differentially expressed genes between the three groups—successful VIT, treatment failure, and healthy controls. Overlapping genes specific for successful VIT with the two transcripts characteristic of successful VIT, regardless of the time point of sampling (From: Gene Cards^®^. The Human Gene Database n.d. Available online: https://www.genecards.org/, accessed on 23 January 2024) (**a**). Overlapping genes specific for treatment failure (**b**). Number of DEGs after VIT in the analysis of successful VIT and healthy controls compared to treatment failure and healthy controls (**c**). (Cut-off: fold change > 5, *p*-value < 0.05.) VIT: venom immunotherapy; DEGs: differentially expressed genes.

**Figure 4 ijms-25-03499-f004:**
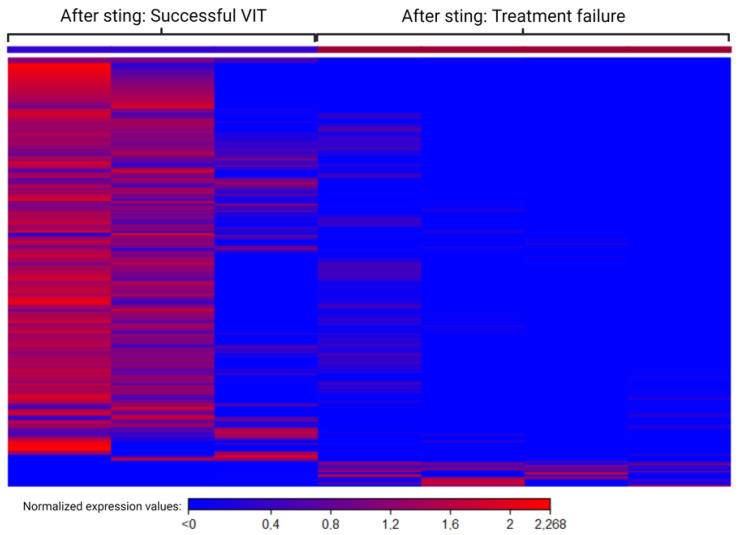
A heat map of normalised expression values of differentially expressed genes after the sting challenge in the groups with successful venom immunotherapy (dark blue) and treatment failure (dark red). Each column represents a patient’s sample; each row represents an individual gene. For each gene, bright blue represents under-expression, and bright red over-expression. (Cut-off: fold change > 5, *p*-value < 0.05.) VIT: venom immunotherapy.

**Figure 5 ijms-25-03499-f005:**
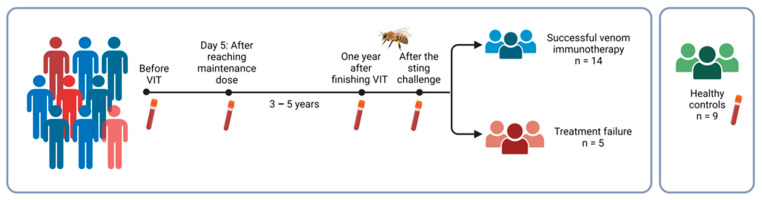
The follow-up study design. Whole-blood samples were collected before the beginning of VIT, after reaching the maintenance dose, one year after finishing VIT, and after a controlled sting challenge. VIT: venom immunotherapy.

**Table 1 ijms-25-03499-t001:** Hub genes differentially expressed after reaching maintenance dose. Positive fold change represents genes upregulated after reaching maintenance dose. Negative fold change represents genes downregulated after reaching maintenance dose.

*Gene*	Ontology [38]	Maintenance Dose vs. Before		Maintenance Dose vs. After	
		Fold Change	*p*-Value	Fold Change	*p*-Value
*IGLV2–5*	Immunoglobulin Lambda Variable 2–5	6.4	0.04	6.5	0.03
*HLA-U*	Major Histocompatibility Complex, Class I, U	6.5	0.04	6.6	0.03
*SERPINB4*	May act as a protease inhibitor to modulate the host immune response against tumour cells	7.2	0.02	7.7	0.01

**Table 2 ijms-25-03499-t002:** Hub genes expressed after reaching maintenance dose and one year after finishing venom immunotherapy. Positive fold change represents genes upregulated after reaching maintenance dose and after venom immunotherapy. Negative fold change represents genes downregulated after reaching maintenance dose and after venom immunotherapy.

*Gene*	Ontology [38]	Maintenance Dose vs. Before		After vs. Before	
		Fold Change	*p*-Value	Fold Change	*p*-Value
*DUX4L10*	Involvement in inflammation, cellular migration, and chemotaxis	−240.1	6.46 × 10^−7^	−7.3	0.0009
*DUX4L12*	Involvement in inflammation, cellular migration, and chemotaxis	−90.1	5.48 × 10^−5^	−6.5	0.002
*CFL1P2*	Required for the upregulation of the atypical chemokine receptor ACKR2	−9.9	0.02	−8.3	0.02
*HTN3*	They function as antimicrobial peptides and are important components of the innate immune system	−7.6	0.009	−11.7	0.002
*CDH9*	Cadherins are calcium-dependent cell adhesion proteins	−7.0	0.008	−5.6	0.01
*PCDHB8*	Calcium-dependent cell adhesion protein involved in cell self-recognition and non-self discrimination	−6.9	0.01	−8.8	0.002
*USP17L22*	Regulates different cellular processes, which may include cell proliferation, progression through the cell cycle, apoptosis, cell migration, and the cellular response to viral infection	−6.5	0.01	−13	0.004
*IKBKGP1*	The NF-kappa-B essential modulator (also known as NEMO) regulating the NFкB pathway, which is involved in many immune and inflammatory responses	−5.5	0.01	−24.9	0.00001
*MAPK15*	Regulates several processes, such as autophagy, protein trafficking, and genome integrity, in a kinase activity-dependent manner	−5	0.03	−5.6	0.01

**Table 3 ijms-25-03499-t003:** Hub genes in the group of patients who tolerated a sting (successful venom immunotherapy; after vs. before). A positive fold change represents genes upregulated after venom immunotherapy. A negative fold change represents genes downregulated after venom immunotherapy.

*Gene*	Ontology [38]	Fold Change	*p*-Value
*PHBP6*	Interacts with STAT3 to affect IL17 secretion in T-helper Th17 cells	−24	0.001
*DUX4*	Involvement in inflammation, cellular migration, and chemotaxis	−18.5	0.0007
*ZG16*	Predicted to act upstream of or in defence response to Gram-positive bacteria	−12	0.007
*CACNG5*	Involved in TCR signalling	−10.4	0.01
*CPN1*	Among its related pathways are the complement cascade and innate immune system	−10	0.011
*TMEM196*	Predicted to be an integral component of the membrane. Diseases associated with TMEM196 include Adult-Onset Severe Asthma	−9	0.002
*IGSF21*	Proteins in this superfamily are usually found on or in cell membranes and act as receptors in immune response pathways	−9	0.007
*DUX4L14*	Involvement in inflammation, cellular migration, and chemotaxis	−8.3	0.03
*TDO2*	This gene encodes a heme enzyme that plays a critical role in tryptophan metabolism	−8	0.004
*MMD2*	Monocyte To Macrophage Differentiation Associated 2; annotations related to this gene include protein kinase activity	−8	0.026
*TWIST1*	Represses expression of pro-inflammatory cytokines such as TNFA and IL1B	−7	0.014
*DUX4L2*	Involvement in inflammation, cellular migration, and chemotaxis	−6.5	0.01
*UMOD*	May serve as a receptor for the binding and endocytosis of cytokines (IL-1, IL-2) and TNF	−6	0.043
*DPT*	Dermatopontin is postulated to modify the behaviour of TGF-beta through interaction with decorin. Enhances TGFB1 activity.	−6	0.014
*SLC17A2*	Predicted to be located in the membrane. Predicted to be active in the lysosome.	−6	0.018
*GNB3*	Related to chronic rhinosinusitis.	−6	0.006
*PLA2G10*	This gene encodes a member of the phospholipase A2 family of proteins. This enzyme plays a role in the production of various inflammatory lipid mediators, such as prostaglandins.	−5	0.016
*NLRP8*	NLRP genes play roles in the mammalian innate immune system through inflammasome formation and the activation of caspases	−5	0.042
*EBI3*	It encodes a secreted glycoprotein and forms interleukin 27 (IL-27). IL-27 partially regulates T-cell and inflammatory responses by activating the Jak/STAT pathway of CD4+ T cells. Among its related pathways are IL27-mediated signalling events and interleukin-12 family signalling	−5	0.040
*PCDHA9*	Potential calcium-dependent cell adhesion protein. It may be involved in establishing and maintaining specific neuronal connections in the brain	−5	0.013
*IGKV2D-28*	The V region of the variable domain of immunoglobulin light chains that participate in the antigen recognition	6	0.002
*E2F4P1*	It plays a vital role in suppressing proliferation-associated genes, and its gene mutation and increased expression may be associated with human cancer	7	0.042

**Table 4 ijms-25-03499-t004:** Study group characteristics.

	Patients on VIT		Healthy Controls	*p*-Value
	Successful VIT	Unsuccessful VIT		
No. of patients	14	5	9	/
Age [years], median (range)	50 (27–63)	49 (26–59)	47 (24–58)	ns ^1^
Sex, no. of patients (%) Female Male	3 (21%) 11 (79%)	0 5 (100%)	2 (22%) 7 (78%)	ns
Specific honeybee IgE [kIU/L], median (range) ^2^	5.48 (0.45–52.40)	3.46 (0.82–11.4)	/	ns
Double sensitisation; honeybee and wasp sIgE, no. of patients (%) ^2^	10 (71%)	1 (20%)	/	ns
Total IgE [IU/mL], median (range) ^2^	113 (3–661)	32 (8–297)	/	ns
Years of VIT, median (range)	6 (6–7)	7 (6–7)	/	ns
Complications during VIT, n (%)	1 (7%)	5 (100%)	/	0.0005

^1^ ns, not significant; ^2^ measurements of specific IgE and total IgE were made at the beginning of venom immunotherapy.

## Data Availability

Data are contained within the article and Appendix A. Any additional supporting data are available from the corresponding author upon reasonable request.

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
