# Peer review of "Blood Transcriptomics Identifies Multiple Gene Expression Pathways Associated with the Clinical Efficacy of Hymenoptera Venom Immunotherapy"

_ijms, 2024, doi:10.3390/ijms25063499_

Round 1

Reviewer 1 Report

Comments and Suggestions for Authors

The manuscript presents very interesting data and a very interesting scientific discussion concerning the application of blood transcriptomics to identify the molecular pathways associated to the efficacy of Hymenoptera venom immunotherapy. I have only minor comments.

The paper uses to many abbreviations. I suggest evaluating which abbreviations are relevant or not. Even for some abbreviations, sometimes it is also described in extension.

In the abstract, I suggest removing the methods used to conduct RNA sequencing and the bioinformatics analysis.

At the end of the introduction, the results of the current work should be eliminated, since they were already pointed in the abstract. I give the congratulations, for the nice introduction section, by  the comparison of studies already conducted, and pointing the  different approach consucted in the present work.

It is not clear in the Materials and Methods Section, if the transcriptomics was conducted on:

- serum, after cells elimination

- or were from whole blood, including cell lysis?

- or were from cells (after e.g. centrifugation) (and if so, what type of  cells)

Pg 111, I do not understand what the p-value  is concerning? I do not thing that it is nnecessary any  statistical evaluation here.

Legend of Fig 1, is highly incomplete, for example should include  what means the “before” and “after”, “maintenance or without the maintenance”, etc. Must also include the criteria to chose these genes, the p<0.05 and 5 fold changes.

Comments on the Quality of English Language

The paper uses to many abbreviations. I suggest evaluating which abbreviations are relevant or not. Even for some abbreviations, sometimes it is also described in extension.

Reviewer 2 Report

Comments and Suggestions for Authors

Dear Editor,

Luzar and collegues presented a study on allergen-specific venom immunotherapy (VIT) to identify molecular mechanisms associated with VIT
efficiency. They performed RNA-sequencing of whole blood and identified a number of target transcripts and associated pathways, which seem to play a role in VIT.

The authors present a number of identified transcripts, which they analyzed for their relevance with regard to the efficiency of a VIT. However, several details of the study could be improved.

1) The group sizes differ greatly, therefore the number of patients with unsuccessful VIT and the control group should be increased to a number similar to the group with successful VIT.

2) The authors should show confirmatory PCR analyses for the sequencing results in order to confirm the statements with a second, independent technique.

3) The study is based on the results of RNA sequencing, but in order to better understand the relevance of the results, the relevance of secreted mediators must also be demonstrated, for example in serum. To this end, the authors could carry out an analysis of secreted mediators in their transcripts (see PMID: 26577568) in order to identify selected targets for this review.

4) In addition, it would be interesting to carry out a string analysis, e.g. of particularly highly/lowly expressed genes, in order to better understand the relationships.

5) The listing of allergen-specific IgEs and their cross-reactivities are also of particular interest and should be shown (PMID: 36440491). In addition, the characterization of the patients must be expanded as far as possible. Also e.g. total IgE, IgG4 and IgA would be beneficial for the readership and a better understanding.

Comments on the Quality of English Language

Please recheck spelling and interpunctation.

Reviewer 3 Report

Comments and Suggestions for Authors

The paper by Luzar et al was a transcriptomic analysis of gene expression profiles of patients undergoing venom immunotherapy.  The authors explored gene expression in healthy controls versus patients at various stages of treatment and with different outcomes.  The authors were able to identify markers of success and failure and they attempt to understand these identified markers in terms of known literature of immune system function.  Overall the paper is well executed and well performed.  A few points to consider could improve the manuscript in presentation and discussion.

Lines 226-227.  It is worth commenting more on the humoral or B-cell pathways that are affected.  Is this possibly related to the changing isotypes during immunotherapy?

Lines 237-239 and accompanying figure.  If this is the main point, the figure focuses more on the venn diagram.  The key summary of each is somewhat hidden in the legends outside the venn diagram.  Strongly consider adding a bar graph of the totals from the 4 categories as an additional panel that would communicate this point better.

Section 2.8 The other sections had nice detailed comparisons of the groups.  Was this not as important?

Figure 4 is hard to understand.  The color bar and the two different patient colors are too similar for easy understanding.   Consider contrasting colors or just braces and labels for the two groups, or separate graphs.  The color bar is not labeled with units.  The legend for Successful versus unsuccessful VIT should be placed closer to the bar for which these labels are important, in the current version it should be at the top.

Discussion last sentence:

if you had a fine grained longitudinal study, do you think you could pinpoint time points and markers that are crucial, and might be useful for customizing the length of therapy for patients?

It would be interested to learn if these markers or patterns correlate with any studies of therapies to airborne or oral allergens.  Clearly this is a large topic but maybe pointing out a few possible correlations to other studies could be useful.

Reviewer 4 Report

Comments and Suggestions for Authors

The paper focuses identification of transcriptomics features associated with the clinical efficacy of VIT. The scientific relevance and interest lies in conducting paired analyses at specific time points - from before the start of VIT treatment, to reaching the maintenance dose, to performing the sting challenge. A limitation of the study is the small study group (19 patients treated with VIT and 5 controls).

The present paper is informative and systematic. However, the Authors should address the following issues to improve the quality of the article.

1.       Lines 96-105: The paragraph is more appropriate for the conclusion than the introduction.

2.       Line 110: Instead of: "had complications during VIT" it should be "had complications during sting challenge".

3.       Lines 110-111: It should be explained what is meant by "treatment success" and "treatment failure".

4.       Discussion is rather limited. The importance of the transcriptomic features shown in the mechanism of VIT should be better explained in this part.
